# OpenReview forum: "ToxicChat: Unveiling Hidden Challenges of Toxicity Detection in Real-World User-AI Conversation"
_EMNLP/2023/Conference — EMNLP 2023 Findings_

### Official Review · Reviewer_v1Tj · 2023-07-23

**Typos Grammar Style And Presentation Improvements:** The accuracy of AHSD in Table 1 shoul…
**Soundness:** 2

**Excitement:**

2: Mediocre: This paper makes marginal contributions (vs non-contemporaneous work), so I would rather not see it in the conference.

**Paper Topic And Main Contributions:**

This paper introduces TOXICCHAT, a benchmark for toxicity detection collected from real-world human-AI interactions. The authors use a human-AI collaborative annotation framework to label the prompt as toxic, non-toxic, or uncertain. Furthermore, the "jailbreaking" prompts are especially labeled. The authors conduct experiments on several existing toxicity classifiers. Experiment results show that these classifiers perform unsatisfactorily on TOXICCHAT. And fine-tuning on TOXICCHAT can improve the performance on the collected real-world validation set.



**Questions For The Authors:**

A. Are the chatbot outputs considered in Table 1 and Table 2?
B. The combination results in Table 2 are unsatisfactory. So how can we leverage the proposed dataset in real-world scenarios to detect toxic language from diverse domains?

**Reasons To Accept:**

1. Toxic prompts (e.g., "jailbreaking" prompts") for recent instruction-tuned LLMs during human-AI interactions can be different from those in previous toxic benchmarks, which are mostly collected from social media. Therefore, it is reasonable to collect these new types of toxic prompts during human-AI interactions.

**Reasons To Reject:**

1. Lack of more in-depth analysis of the collected toxic prompts. For example, there are diverse fine-grained types of jailbreak prompts and failure modes of LLMs ([1]).
2. The motivation and writing are confusing. While the title mentions user-AI interaction, the paper mainly focuses on detecting toxicity in prompts alone. Chatbot outputs are not considered until Section 4.2. However, detecting toxic prompts to LLMs during human-AI interaction should be different from detecting toxic statements. Therefore, I would recommend polishing the writing and adding experiments on previous dialogue safety benchmarks ([2][3]).
3. Lack of experiments with stronger toxicity detection methods. The toxicity detection models used in current experiments are either open-source APIs (which are known to have low performance in the toxicity detection task) or fine-tuned RoBERTa-base models. However, recent works have shown the strong evaluation abilities of LLMs, surpassing previous SOTA baselines on various tasks, including toxicity detection ([4][5]).
4. While the most interesting part of the collected dataset is the "jailbreaking" prompts, it only accounts for 3.89% of the dataset, i.e., about 110 examples.


[1] Jailbroken: How Does LLM Safety Training Fail?
[2] Bot-Adversarial Dialogue for Safe Conversational Agents
[3] On the Safety of Conversational Models: Taxonomy, Dataset, and Benchmark
[4] G-Eval: NLG Evaluation using GPT-4 with Better Human Alignment
[5] Is ChatGPT better than Human Annotators? Potential and Limitations of ChatGPT in Explaining Implicit Hate Speech


**Reproducibility:**

3: Could reproduce the results with some difficulty. The settings of parameters are underspecified or subjectively determined; the training/evaluation data are not widely available.

**Reviewer Confidence:**

5: Positive that my evaluation is correct. I read the paper very carefully and I am very familiar with related work.

---

> ### Author Rebuttal · Authors · 2023-08-29
>
> Dear Reviewer v1Tj,
>
> Thanks for your comments. We hope our response can address your concerns.
>
> > In-depth analysis of the collected toxic prompts
>
> Our study adheres to the definition of jailbreaking as the act of tricking or guiding the chatbot to provide outputs that are intended to be restricted by internal governance and ethics. We recognize the need for a more detailed examination and will incorporate a comprehensive analysis in future work.
> During our investigation, we also explore several specific types of user prompts that are worth noting. For example, in prompts where users ask the model for role-playing, we discovered that 23.32% of these interactions manifested toxic behavior. This suggests a pronounced propensity for toxicity within role-playing prompts. We will include the analysis in the camera ready.
>
> > Regarding the motivation and writing
>
> Thank you for your feedback. We focus on detecting toxicity in prompts in our paper. The reasons for this focus are:
> - While there is existing work on addressing toxicity in machine-generated content, such as Toxigen, the toxicity in user prompts during real-world user-AI interactions remains less explored.
> - Given the rising prominence of platforms like ChatGPT, we believe it's vital to first moderate user prompts to ensure subsequent generated content is within acceptable bounds.
>
> We appreciate the your feedback and will work on refining our writing and incorporating more experiments on earlier dialogue safety benchmarks in future iterations.
>
> > Regarding experiments with stronger toxicity detection methods
>
> The primary objective of our research is not to develop a state-of-the-art toxicity detection model. Our main aim lies in the creation of ToxicChat, which serves as a foundation for training toxicity detection models tailored to the nuances of user-AI interactive environments. It's important to understand that the domain of toxicity detection in user-AI interactions is distinct and presents unique challenges, and our experiments highlight this distinction.
>
> As demonstrated in Table 2, existing works fall short in delivering satisfactory results in this emerging domain. Further, to bridge this knowledge gap, Table 3 provides insights into the salient features that ought to be considered when developing a toxicity detection model specific to this domain. Thus, our work paves the way for future researchers to build upon ToxicChat and align their models to the demands of user-AI interactions more effectively.
>
> > Regarding jailbreaking
>
> Given the natural proportion of jailbreaking examples, it's not feasible to include a vast number of toxic or jailbreaking examples while also maintaining the actual distribution we observe in real-world data. Meanwhile, as reflected in the experimental results shown in Table 2, training on the current dataset size has proven to be adequate, achieving good moderation results.
>
> To answer your questions:
> > Q: Are the chatbot outputs considered in Table 1 and Table 2?
>
> Chatbot outputs are not considered in Table 1 and Table 2.
>
> > Q: The combination results in Table 2 are unsatisfactory. So how can we leverage the proposed dataset in real-world scenarios to detect toxic language from diverse domains?
>
> Thanks for your consideration about the usage of this benchmark. The combination set in Table 2 is a collection of all mentioned existing related benchmarks. The result shows that merely merging existing datasets fails to address the intricate and multifaceted challenges associated with the identification of toxic content in real-world user-AI interactions. While it is a very inspiring future direction about how to utilize ToxicChat in different domains, our primary purpose in constructing such a dataset is to meet the demand for more diverse and nuanced toxicity benchmarks as LLM-based chatbots are widely used.
>
> Also thanks for pointing out the typo issue!

---

### Official Review · Reviewer_vR6f · 2023-08-04

**Soundness:** 2

**Excitement:**

3: Ambivalent: It has merits (e.g., it reports state-of-the-art results, the idea is nice), but there are key weaknesses (e.g., it describes incremental work), and it can significantly benefit from another round of revision. However, I won't object to accepting it if my co-reviewers champion it.

**Missing References:**

[1] Curry, Amanda Cercas, Gavin Abercrombie, and Verena Rieser. "ConvAbuse: Data, Analysis, and Benchmarks for Nuanced Abuse Detection in Conversational AI." Proceedings of the 2021 Conference on Empirical Methods in Natural Language Processing. 2021.


**Paper Topic And Main Contributions:**

This paper presents a new benchmark dataset called ToxicChat which has toxic or abusive conversations with AI agents. The dataset was created based on output from the Vicuna LLM, where the annotation entailed manual annotation of items that were first filtered based on the Perspective API’s toxicity score. The dataset also includes ‘jailbreaking’ attempts in conversations where chatbot users may use deceptive prompts to bypass guardrails. The authors then test existing toxicity and hate speech classifiers on ToxicChat and find that most existing approaches have low performance, especially in detecting jailbreaking attempts.


**Questions For The Authors:**

QuestionA: The JR metric in Table 2 and 3 seems to be accuracy. What was the F1 score for detecting this?

QuestionB: Is ConvAbuse comparable to your benchmark? How does ToxicChat compare against it, especially how is ToxicChat better?

QuestionC: How would you tackle jailbreaking prompts potentially filtered out by the Perspective API filtering step?

QuestionD: This more for clarifying: What is the ‘data’ that the annotators annotate? Is it just the prompt to Vicuna or also its output? What if it is a multi-turn conversation? This would also be a question for the authors


**Reasons To Accept:**

- This paper tackles an important problem of detecting problematic input to chatbots. This is a timely problem, given the rise and deployment of conversational LLMs.
- The dataset seems quite valuable, also since it includes jailbreaking problems, which to the best of my knowledge, has not been studied in the context of toxicity
- Compares against several baselines for similar tasks and finds the better performance of models trained on ToxicChat


**Reasons To Reject:**

- My biggest critique is that it is not entirely true that there are no conversational abusive language datasets based on AI-human interactions. ConvAbuse [1] does exactly this, though the focus is on the toxic outputs of conversational agents rather than inputs to them. However, it is the benchmark closest to ToxicChat, and I think it is quite crucial to compare the latter with ConvAbuse
- While including the jailbreaking attempts in the dataset is definitely interesting and novel, I am wondering if the Perspective API filters out some of these inputs. These inputs are not conventionally ‘toxic’ in nature and might get excluded in the first filtering step. This would mean that the jailbreaking attempts included in ToxicChat also have some toxicity in them and might not be representative of general jailbreaking attempts, reducing the validity of this type of data in the benchmark.
- How robust are the results in Table 1-3? Are these repeated over a few runs? I would suggest reporting some type of error measures like standard deviations or confidence intervals.
- Hard to say if ToxicChat is only useful for Vicuna or not. We could answer this question if the paper also reported a cross-domain analysis, e.g., using the RoBERTa model trained on ToxicChat to test against Toxigen and RealToxicityPrompts.
- Reproducibility based on the submission. This is not a reason to reject the paper, but it is a weakness. The authors have included their dataset. However, they do not include any code nor mention what settings they used for training the models in Tables 2 and 3 (hyperparameters, runs, etc). Therefore, I cannot adjudicate the reproducibility of these results. But I would reiterate this only affects the reproducibility assessment rather than the overall assessment of the paper since the main contribution is the dataset.


**Reproducibility:**

1: Could not reproduce the results here no matter how hard they tried.

**Reviewer Confidence:**

4: Quite sure. I tried to check the important points carefully. It's unlikely, though conceivable, that I missed something that should affect my ratings.

**Typos Grammar Style And Presentation Improvements:**

The paper is generally well-written and easy to follow, barring a few unclear points and typos. The unclear points were:
- ‘JR’ is never mentioned in the text but from context I understand these are the jailbreaking prompts. I would suggest making this explicit in the description of Table 2 and Table 3.
- what the 'data' annotated by the human annotators comprised of (see QuestionD)

Typos: ‘performnace’ —> ‘performance’ (line 300), ‘reply’ —> ‘rely’ (line 166)

---

> ### Author Rebuttal · Authors · 2023-08-29
>
> Dear Reviewer vR6f,
>
> Thanks for your detailed observation and questions. We hope our response can address your concerns.
>
> > Including discuss of ConvAbuse
>
> Thanks for bringing up this related work. ConvAbuse indeed also analyzes toxic behaviors of users when interacting with chatbots. However, during their study, many toxicity arose from user abusing the chatbots (Section 5 Data, in their paper), while in our study on user interaction with LLM-based chatbots, we found almost no such cases, and the toxicity was predominantly user saying things explicit toxic, or user trying to trigger the chatbot to be toxic. Therefore, we believe there is still a domain difference between the toxicities in ConvAbuse and toxicities in user chatbot interactions with LLMs.
>
> To illustrate the domain difference, we conducted the same study in Table 2, and trained a roberta-base model on ConvAbuse to test its performance on our labeled ToxicChat (in addition, we ran the model three times with different random seeds and reported the standard deviation). As the table below shows, the ConvAbuse trained model’s performance is still not as good as a roberta-base model fine-tuned on in domain data in ToxicChat, especially on the jailbreaking part.
>
> Therefore, we believe that the main claims in our paper still stand: existing datasets in different domains do not transfer well to toxicity in user and LLM-based chatbot interactions.
>
> benchmark | acc         | pre        | rec        | f1         | jailbreaking |
> |-------------|-------------|------------|------------|------------|--------------|
> | ConvAbuse   | 94.39±0.10 | 77.87±0.42 | 66.83±1.61 | 70.69±1.43   | 16.67±5.00
> | ToxicChat   | 96.05±0.12 | 84.39±0.69 | 81.28±0.89 | **82.74±0.43** | **80.55±2.50**
>
> > Clarification of the Jailbreaking metric, and discussion of whether the collaborative annotation framework would undermine the jailbreaking test set.
>
> Thanks for raising this question.
> Throughout the paper, jailbreaking is treated as a special tag of the toxic user input — if an user prompt is labeled as toxic, we additionally annotate whether it is a jailbreaking example.
> In this sense, the two metrics that we are concerned with are
> - How well a toxicity detection model can detect toxic examples?
> - How well the model works on the jailbreaking subset of toxicity examples?
>
> Therefore, the metric we are using for jailbreaking is the recall on the annotated jailbreaking examples; the precision is guaranteed to be low since models also need to annotate non-jailbreaking toxic examples, so it is not meaningful. The recall is also equivalent to the accuracy of the jailbreaking examples.
>
> Because of our focus on toxic jailbreaking prompts, the general jailbreaking prompts in the dataset that may bias the perspective api filtering shall not bias our jailbreaking metric.
>
> > Robustness and reproducibility of experimental results.
>
> We conduct a re-evaluation of all experiments in the paper with 3 independent runs with different random seeds, and report the numbers along with standard deviation below. Overall, we find randomness brings in only small variations, and all of the experimental results still support the claims in the paper.
>
> For training, we used standard training scripts from huggingface to train the roberta-base model. We did not tune any hyperparameters as we find them usually non-sensitive. For all experiments, we sticked with the same training recipe, with 5 epochs and a learning rate of 1e-5. We will release the code to train the models and the model weights for reproducibility.
>
> | Domain          | acc        | pre        | rec        | f1         | jailbreaking |
> |-----------------|------------|------------|------------|------------|--------------|
> | RealToxicPrompts | 91.82±0.15 | 69.72±0.10 | 81.32±1.27 | 73.76±0.25 | 35.55±2.50   |
> | Toxigen         | 85.70±2.89 | 59.71±0.69 | 69.61±3.70 | 61.64±0.52 | 81.67±13.33  |
> | Jigsaw          | 94.28±0.08 | 82.42±1.87 | 58.99±0.16 | 63.48±0.40 | 6.11±0.96     |
> | MovieReview     | 93.61±0.00      | 46.80±0.00      | 50.00±0.00      | 48.3±0.00       | 0.00 ±0.00         |
> | HSTA            | 90.07±0.57 | 57.27±1.55 | 56.67±1.09 | 56.98±1.30 | 3.33±0.02     |
> | Combinations    | 93.44±0.19 | 72.01±0.90 | 66.65±0.37 | 68.86±0.40 | 8.33±1.65     |
> | ToxicChat       | 96.05±0.12 | 84.39±0.69 | 81.28±0.89 | **82.74±0.43** | **80.55±2.50**    |
>
> | Features   | acc        | pre        | rec        | f1         | jailbreaking |
> |------------|------------|------------|------------|------------|--------------|
> | In         | 96.09±0.02 | 84.93±0.35 | 80.59±0.97 | 82.58±0.40 | 75.55±4.83   |
> | Out        | 93.61±0.00      | 46.80±0.00      | 50.00±0.00      | 48.35±0.00      | 0.00±0.00        |
> | In ∪ Out   | 96.05±0.12 | 84.39±0.69 | 81.28±0.89 | **82.74±0.43** | **80.55±2.50**   |
>
> > Generalizability of ToxicChat
>
> First, we want to clarify that the aim of this paper is to address the problem of toxicity detection in user-chatbot interactions, and we are analyzing **only** the toxicity of the first-round input of the user to the chatbot, without considering the responses of the chatbot. Therefore, the dataset itself is not related with the responses from Vicuna, and can be applied to more general domains related to user inputs to chatbots. We choose to analyze and annotate only the first round input in particular for this generalizability benefit.
>
> We do not expect the dataset to be useful for a completely different domain (e.g., social media), as we have shown that social media toxicity data does not work well for ToxicChat. It is also not the goal of this paper to create a general toxicity detection model — which we believe would be better done by combining data from different domains and training a model on them.
>
> > About the annotations.
>
> The ‘data’ to be annotated is the user prompt for the user-AI conversation. Our current focus with ToxicChat has been predominantly on the initial turn of the conversation, excluding some non-informative initial prompts like greeting, which makes up about half of the raw conversation data we have collected. In the future, we plan to expand our efforts with ToxicChat to include multi-turn conversations.

---

### Official Review · Reviewer_nmU3 · 2023-08-05

**Soundness:** 3

**Excitement:**

3: Ambivalent: It has merits (e.g., it reports state-of-the-art results, the idea is nice), but there are key weaknesses (e.g., it describes incremental work), and it can significantly benefit from another round of revision. However, I won't object to accepting it if my co-reviewers champion it.

**Paper Topic And Main Contributions:**

In this research, the authors presented TOXICCHAT, a real-world user-AI toxicity detection benchmark consisting of 7,599 user-AI conversations.

**Questions For The Authors:**

- How did the authors decide the size of dataset TOXICCHAT (7,599)?
- TOXICCHAT includes around 483 toxic data points and 297 jailbreaking data points. Do the authors think those numbers are enough in the dataset?

**Reasons To Accept:**

- The authors created a new user-AI toxicity detection benchmark from real world data.
- The authors reduced annotation labor (around 60%) through good API capability analysis.
- The authors showed toxicity/jailbreak detection on TOXICCHAT was difficult for open-sourced APIs, and RoBERTa model trained on TOXICCHAT was better than the other dataset.

**Reasons To Reject:**

- Explanation is limited (in some sections, it is difficult to understand).
- Experiment (Table.2) is not very fair.  The reviewer thinks that the model trained on TOXICCHAT should be better obviously because other datasets may not cover user-AI conversation.

**Reproducibility:**

3: Could reproduce the results with some difficulty. The settings of parameters are underspecified or subjectively determined; the training/evaluation data are not widely available.

**Reviewer Confidence:**

4: Quite sure. I tried to check the important points carefully. It's unlikely, though conceivable, that I missed something that should affect my ratings.

---

> ### Author Rebuttal · Authors · 2023-08-28
>
> Dear Reviewer nmU3,
>
> We really appreciate your comments and your support for our work. We hope our response can address your concerns.
>
> > Explanation is limited (in some sections, it is difficult to understand).
>
> Thanks for your feedback. We will try our best to ensure the content remains understandable to readers. If you could specify which sections you found challenging, we are willing to provide additional insights or consider revising for improved clarity in future iterations.
>
> > Experiment (Table.2) is not very fair. The reviewer thinks that the model trained on TOXICCHAT should be better obviously because other datasets may not cover user-AI conversation.
>
> Indeed, the model trained on TOXICCHAT performed better, which underscores our very intention with the experiment. Our primary goal was to highlight the limitations of existing models when applied to the current user-AI interactive environments, e.g., ChatGPT or Vicuna. By exploring the performance of models trained on other datasets against the TOXICCHAT dataset, we have shown the necessity for specialized training data in achieving satisfactory moderation results in user-AI interactions. We believe this distinction is pivotal in driving the field forward.
>
> > How did the authors decide the size of dataset TOXICCHAT (7,599)?
>
> The dataset's size was determined by two primary considerations: (1) Perspective API's error tolerance rate and (2) the capacity of our annotators to manage their workload. As shown in Figure 2, our goal was to focus on a probability range where the error rate is below 2%, which means we need to annotate about 40% of the entire data. Based on the trial data annotations from Section 2.1, and to strike the right balance between the timeline and workload, we chose to annotate roughly 3000 examples, setting aside a portion for exclusion in case some of the data is deemed unsuitable for data release (less than 100 examples).
>
> > TOXICCHAT includes around 483 toxic data points and 297 jailbreaking data points. Do the authors think those numbers are enough in the dataset?
>
> According to our exploration, in our real-world chat data, the occurrence rate of toxic content is about 6%, and jailbreaking is about 3%. Given these proportions, it's not feasible to include a vast number of toxic or jailbreaking examples while also maintaining the actual distribution we observe in real-world data. Meanwhile, as reflected in the experimental results shown in Table 2, training on the current dataset size has proven to be adequate, which achieves good moderation results.

---

### Meta-Review · Area_Chair_UCNy · 2023-09-18

**Recommendation:** 3

**Metareview:**

The authors present a new benchmark dataset that is built from human-chatbot interactions, which are different from the social media data that many toxicity detectors are trained on. The reviewers agree that this is an important problem in detecting toxic inputs to chatbots. They agree mostly that the dataset will be valuable and includes interesting jailbreaking prompts. They also show that models perform better when trained on their dataset. Although there is interest in the jailbreaking prompts, the reviewers note that they account for less than 4% of the data. They also note a lack of comparison to stronger models. They also note some previous work such as ConvAbuse that should be mentioned in the paper. Many of the smaller concerns have been addressed by the authors in their rebuttal including the analysis of prompts, robustness, and discussion of ConvAbuse or can be addressed in the camera ready, such as the writing improvements.

---

### Decision · Program_Chairs · 2023-10-07

**Decision:**

Accept-Findings

**Comment:**

The authors present a new benchmark dataset that is built from human-chatbot interactions, which are different from the social media data that many toxicity detectors are trained on. The reviewers agree that this is an important problem in detecting toxic inputs to chatbots. They agree mostly that the dataset will be valuable and includes interesting jailbreaking prompts. They also show that models perform better when trained on their dataset. Although there is interest in the jailbreaking prompts, the reviewers note that they account for less than 4% of the data. They also note a lack of comparison to stronger models. They also note some previous work such as ConvAbuse that should be mentioned in the paper. Many of the smaller concerns have been addressed by the authors in their rebuttal including the analysis of prompts, robustness, and discussion of ConvAbuse or can be addressed in the camera ready, such as the writing improvements.